# Landscape diversity and local temperature, but not climate, affect arthropod predation among habitat types

**Ute Fricke**[1]*, **Ingolf Steffan-Dewenter**[1], **Jie Zhang**[1], **Cynthia Tobisch**[2,3], **Sandra Rojas-Botero**[3], **Caryl S. Benjamin**[4], **Jana Englmeier**[5], **Cristina Ganuza**[1], **Maria Haensel**[6], **Rebekka Riebl**[6], **Johannes Uhler**[5], **Lars Uphus**[4], **Jörg Ewald**[2], **Johannes Kollmann**[3], **Sarah Redlich**[1]

1 Department of Animal Ecology and Tropical Biology, Biocenter, Julius-Maximilians-University Würzburg, Würzburg, Germany, 2 Institute for Ecology and Landscape, Weihenstephan-Triesdorf University of Applied Sciences, Freising, Germany, 3 Restoration Ecology, TUM School of Life Sciences, Technical University of Munich, Freising, Germany, 4 Ecoclimatology, TUM School of Life Sciences, Technical University of Munich, Freising, Germany, 5 Field Station Fabrikschleichach, Department of Animal Ecology and Tropical Biology, Julius-Maximilians-University Würzburg, Rauhenebrach, Germany, 6 Professorship of Ecological Services, Bayreuth Center of Ecology and Environmental Research (BayCEER), University of Bayreuth, Bayreuth, Germany

\* ute.fricke@uni-wuerzburg.de

**Data Availability Statement:** Data are available in the Dryad Digital Repository: https://doi.org/10.5061/dryad.f1vhhmh04 (Fricke et al, 2022) R code

## Abstract

Arthropod predators are important for ecosystem functioning by providing top-down regulation of insect herbivores. As predator communities and activity are influenced by biotic and abiotic factors on different spatial scales, the strength of top-down regulation ('arthropod predation') is also likely to vary. Understanding the combined effects of potential drivers on arthropod predation is urgently needed with regard to anthropogenic climate and land-use change. In a large-scale study, we recorded arthropod predation rates using artificial caterpillars on 113 plots of open herbaceous vegetation embedded in contrasting habitat types (forest, grassland, arable field, settlement) along climate and land-use gradients in Bavaria, Germany. As potential drivers we included habitat characteristics (habitat type, plant species richness, local mean temperature and mean relative humidity during artificial caterpillar exposure), landscape diversity (0.5–3.0-km, six scales), climate (multi-annual mean temperature, 'MAT') and interactive effects of habitat type with other drivers. We observed no substantial differences in arthropod predation rates between the studied habitat types, related to plant species richness and across the Bavarian-wide climatic gradient, but predation was limited when local mean temperatures were low and tended to decrease towards higher relative humidity. Arthropod predation rates increased towards more diverse landscapes at a 2-km scale. Interactive effects of habitat type with local weather conditions, plant species richness, landscape diversity and MAT were not observed. We conclude that landscape diversity favours high arthropod predation rates in open herbaceous vegetation independent of the dominant habitat in the vicinity. This finding may be harnessed to improve top-down control of herbivores, e.g. agricultural pests, but further research is needed for more specific recommendations on landscape management. The absence of

is available at Zenodo: https://doi.org/10.5281/zenodo.6467203 (Fricke et al, 2022)

**Funding:** This study was conducted within the LandKlif project funded by the Bavarian Ministry of Science and the Arts via the Bavarian Climate Research Network (bayklif, www.bayklif.de). This publication was supported by the Open Access Publication Fund of the University of Wuerzburg.

**Competing interests:** The authors have declared that no competing interests exist.

MAT effects suggests that high predation rates may occur independent of moderate increases of MAT in the near future.

## Introduction

Predation and parasitism are frequent causes of mortality to many herbivorous insect species [1] and hence can exert strong impact on herbivore communities [2, 3]. Through their impact on herbivores, natural enemies can also indirectly affect plant damage, vegetation structure and composition, and nutrient cycling [2–4]. This renders natural enemies and their biotic interactions essential to ecosystem functioning. Important natural enemies for the regulation of herbivorous insects are arthropod predators [5]. Predation intensity can differ between habitat types [6, 7], albeit direct comparisons among typical habitat types in temperate regions (forest, grassland, arable fields and settlements) are lacking. Besides, arthropod activity is influenced by local weather conditions [8], while plant species richness [9], climate [10] and regional land use [11] affect arthropod communities, with potential consequences for top-down suppression of herbivores [12, 13]. However, the combined effects of these drivers on arthropod predation in different habitats are largely unknown, albeit urgently needed with regard to anthropogenic climate and land-use change.

Local habitat characteristics such as habitat type, plant species richness and weather conditions affect predator richness, activity or both with possible implications on predation rates. With respect to habitat type, Ferrante et al. [7] observed higher predation rates in forests than in maize fields. This may be related to on average higher natural enemy richness in natural than agricultural ecosystems [14], which possibly translates into higher and lower predation rates, respectively [12]. Plant species richness was described both to benefit natural enemies [15] and predation rates [16], whereby predation rates may be affected directly or indirectly, via changes in the composition of the natural enemy community [12, 14]. Besides, higher plant species richness can also lead to higher structural complexity of the vegetation [9], which may alter predator behaviour with positive effects on predation rates, e.g. reduced intraguild predation [17]. However, knowledge of plant richness effects on natural enemies and their services originate almost exclusively from plant diversity experiments [e.g. 9, 15, 16], while complementing field studies are lacking. Similar applies to weather conditions. Temperature and humidity modify arthropod activity in terms of catchability by traps [8], but little is known about their effects on predation rates. For instance, activity of predatory carabid beetles increases with temperature [8, 18], and, depending on species traits, decreases towards higher relative humidity [8], while too low temperatures can restrict carabid activity [19]. Thus, the richness and activity of predators are affected through habitat characteristics, yet the consequences for predation rates are much less clear.

At a regional scale, landscape complexity and climatic factors impact predators. In complex landscapes, both species richness and abundance of generalist enemies are higher, and top-down control of herbivorous arthropods is commonly increased [11, 20, 21]. Considering landscape diversity as an aspect of landscape complexity [20], diverse landscapes can provide complementary or supplementary resources to organisms moving between habitat patches with beneficial effects on their population size [22]. Thus, predation rates may increase towards more diverse landscapes. In addition, climate change, and in particular a warmer climate, is expected to affect arthropods in many aspects, for instance, in their geographic distribution and life history traits [10, 23]. Consequently, this may impact predation rates. Indeed,

the efficacy of predators to suppress herbivores can increase with mean annual temperatures [24] as well as predation rates increase towards lower altitudes and latitudes [25]. Thus, both diverse landscapes and warm climates may favour higher predation rates.

Here we use arthropod attack marks on artificial caterpillars, facilitating standardized estimates of predation rates over large spatial scales, to study the combined effects of local habitat type, plant species richness, weather, landscape diversity and multi-annual mean temperature on arthropod predation, and ask whether effects differ among habitat types. This study advances the understanding of top-down regulation of herbivores and natural pest control services in the context of climate and land use.

## Material and methods

### Study area and plot selection

This study was conducted within the LandKlif project in Bavaria, Germany, which used a novel multi-scale study design to disentangle the combined effects of climate and land use on biodiversity and ecosystem functions [26], here on arthropod predation rates. From grid cells (5.8 km x 5.8 km) covering Bavaria (Germany), 60 grid cells (= 60 'regions') were selected encompassing four replicates of 15 combinations of climate zones (multi-annual mean temperature between 1981–2010; < 7.5°C, in 0.5°C steps until 9°C, > 9°C) and landscape-scale land-use types (near-natural, agriculture and urban) (S1 Fig). Regional land-use types were defined as near-natural when > 85% of the region were covered by near-natural vegetation with a minimum of 50% forest, as agriculture when > 40% were covered by arable land and managed grassland, and as urban when > 14% were covered by housing, industry and traffic infrastructure. In each selected region, plots were established in the three habitat types dominating in the respective region (out of four possible types: forest, grassland, arable field or settlement), while avoiding overlap of 1-km 'buffer zones' among plots. When habitat types happened to be similarly represented in a region, the more contrasting habitat type was chosen. This means, if arable field and settlement had been selected by dominance, forest rather than grassland was chosen as third habitat type. Plots were installed as 30 m x 3 m experimental strips on open herbaceous vegetation–e.g. forest glades, grassland, field margins, parks– with at least 50 m distance to larger roads and other habitat types (e.g. between forest and an experimental strip on a field margin). Land owners (e.g. private individuals) or their official representatives (i.e. in case of ownership through municipalities, church, the Bavarian State Forest and the National Park Bavarian Forest) approved access to their land and the conduct of experiments, as well as the leaseholders, where applicable. Research on predation rates was realised on 147 out of 179 LandKlif plots, yet complete data sets were acquired for 113 plots (data exclusion criteria, see below).

### Predation rate assessment

Arthropod predation rates were assessed using standardised green artificial caterpillars (diameter 3 mm; length 20 mm) made from plasticine (Weible Fantasia KNET grün, Weible GmbH & Co. KG, Germany), as suggested for rapid ecosystem function assessment with large geographic extent [27]. Brown pieces of paper (size 40 x 19 mm; 100 g m$^{-2}$, hazelnut brown, paper type "Paperado", Rössler Papier GmbH & Co. KG, Germany) served as carrier onto which the artificial caterpillars were glued (UHU Alleskleber extra tropffrei gel, UHU GmbH & Co. KG, Germany); 20 artificial caterpillars were placed on every plot at ground level below vegetation but above litter to standardize position. The artificial caterpillars were spread across the 30 m x 3 m experimental strip with at least 1-m spacing between two caterpillars and other experimental items (e.g. Malaise trap). Bamboo sticks with a red tip were punched through a hole in

the paper carrier to fix and mark the position of each artificial caterpillar. The collection of the caterpillars started after 48 ± 6 hours (range: 42–54 hours). The presence or absence of arthropod attack marks was assessed in the field using reference images provided by Low et al. [28]. Arthropod attack marks were not further differentiated into finer taxonomic level as this is error prone and hence not recommended [28]. We calculated predation rates per plot as the proportion of artificial caterpillars with arthropod attack marks after 2-d exposure relative to the total number of caterpillars per site. We call the obtained measure 'predation rate', as ground-active arthropods, particularly carabids (Coleoptera: Carabidae), are among the most frequent attackers of artificial caterpillars at ground-level and as attack marks of parasitoids are rare [16, Personal observation UF]. Arthropod predation rates were assessed once per plot in May (starting dates between 10[th] and 25[th] May 2019).

## Measures of habitat characteristics

Plots were established in different local habitat types (forest, grassland, arable field and settlement). Through establishing plots in forest glades, extensive grasslands, crop field margins and green areas in settlements within the different local habitat types, exposure of artificial caterpillars was standardized to open herbaceous vegetation.

Plant species richness per plot was derived between May and July 2019 from plant species records in seven subplots (10 m$^2$ total sampling area). Further details and a species list are provided in Fricke et al. [29].

Local weather conditions during caterpillar exposure were derived from thermologgers (ibutton, type DS1923). Those were attached north-facing to a wooden pole, at 1.1 m above ground and roughly 0.15 m below a wooden roof, which prevented direct solar radiation. One thermologger was established per plot. We extracted mean temperature and mean relative humidity (in the following referred to as 'local mean temperature' and 'mean relative humidity') during the study-site specific exposure period of the artificial caterpillars from hourly measurements of the thermologgers.

## Measures of regional land use and climate

Landscape diversity was calculated as Shannon Index from detailed land-cover maps (combination of ATKIS 2019, CORINE 2018 and IACS 2019, see [29]) based on six main land-cover types (semi-natural habitat, forest, grassland, arable, urban, water). Thus, high landscape diversity indicates more different land-cover classes, more similar proportions of them or both. Landscape diversity was calculated in radii around the centre point of the plots at six spatial scales (0.5–3.0 km, in 500-m steps). At 2-km scale, low landscape diversity equated a dominance of forest or arable land, and the land-cover proportions of semi-natural habitat and water were below 7.5% and 10.2%, respectively.

We retrieved 30-year multi-annual mean temperatures (1981–2010, MAT) per plot based on gridded (1-km resolution) monthly averaged mean daily air temperatures [30].

## Data analysis

Prior to data analysis, data exclusion criteria were applied to standardize data. We excluded artificial caterpillars exposed to attack for more than 54 hours (exceeding 48 ± 6 h limit), 'released' later than 25[th] May, and recovered incomplete with a loss of more than 20% (<16 artificial caterpillars per plot). In total, we achieved standardized data on 113 plots. Artificial caterpillars from 58 of these plots (51%) were transported to the lab to double-check the assessments done in the field. Field and lab assessments of arthropod predation rates were positively

correlated (Pearson's r = 0.79; S2 Fig). In the following, arthropod predation rates refer to the field observations (113 plots).

Arthropod predation rate data were analysed with binomial generalized mixed effect models to cope with proportional data (derived from absence-presence data) using the R-package 'glmmTMB' [31] with R version 4.0.3 [32]. Region was included as a random term to account for the nested study design and was retrieved throughout the model selection process [33]. Due to zero-inflation (complete absence of attack from 17% of plots), confirmed using the R-package 'DHARMa' [34], we added a zero-inflation term. We did not account for exposure duration of the artificial caterpillars in the models, since data were standardized by exposure duration (48 ± 6 h limit) and similar exposure durations of 48.2 ± 1.7 h (mean ± SD) were realized among plots.

As candidate predictors (fixed effects) of arthropod predation rates, we included habitat type, plant species richness, local mean temperature and mean relative humidity (during artificial caterpillar exposure), landscape diversity and MAT. Candidate predictors were z-transformed prior to analysis, while presented models contain untransformed predictor variables.

To parametrize the zero-inflation term, we considered factors which might explain absence of attack on plot level, e.g. arthropod activity limited by low temperatures [19]. Besides, we visually screened the candidate predictors for accumulation of absence-of-attack events (predation rate = 0) at the extremes of the predictor ranges. Local mean temperature was the only candidate predictor in which absence of attack marks was frequently observed at the lower range on a per plot basis. Therefore, local mean temperature was included as a single candidate predictor in the zero-inflation term. Additionally, we run a separate analysis on presence-absence of attack on plot level (data extracted from predation rate data; predation rate > 0 replaced by 1) to investigate how the probability of attack on plot level was affected through local mean temperature using binomial generalized linear mixed effect models including region as random term (see S1 Table).

When analysing the data, we first conducted multi-model averaging to identify the most relevant predictors and spatial scales. Models with all possible predictor combinations were created separately for each spatial scale (0.5–3.0 km, six scales). Akaike weights were computed using the dredge-function from the 'MuMin' R-package [35]. Achieved Akaike weights ($w_i$) were summed per predictor and spatial scale, whereby high summed Akaike weights ($\Sigma w_i$; range: 0 (low)– 1 (high)) indicate a high relative importance of a predictor, corresponding to high cumulative probability that a predictor occurs in the best model at the respective spatial scale [36].

In a second step, we analysed potential interactive effects of habitat type with plant species richness, weather conditions during artificial caterpillar exposure (local mean temperature, mean relative humidity), landscape diversity and MAT. Therefore, we added single interaction terms (e.g. local habitat type * plant species richness) to the best model at the most relevant spatial scale derived from multi-model averaging. Model selection was done based on Akaike's information criterion corrected for small sample size (AICc). Thereby, models with lower AICc were considered better, and models with ΔAICc < 2 were considered equal and the more parsimonious model was chosen.

Pearson correlations between continuous candidate predictor variables were rather low ≤ 0.33 (S2 Table) with two exceptions. MAT was positively correlated with local mean temperature (Pearson's r = 0.59) and negatively correlated with mean relative humidity (Pearson's r = -0.51). However, all variance inflation factors (VIF) fell below the commonly applied threshold for collinearity of variance inflation factor >10 (30, see S3 Table), unless interactions with the only categorical variable habitat type were included (S4 Table), which commonly inflates VIF; the latter were calculated using the R-package 'performance' [37]. Local mean

temperature (Kruskal Wallis, $P = 0.070$), mean relative humidity (Kruskal Wallis, $P = 0.219$) and landscape diversity (2-km scale, Kruskal Wallis, $P = 0.187$) were similar among habitat types whereas plant species richness was higher in grasslands than arable plots and intermediate in forests and settlements (Kruskal Wallis, $P = 0.022$; Bonferroni-corrected Wilcoxon test), and MAT was higher in settlements than forests and grasslands, and intermediate in arable plots (Kruskal Wallis, $P = 0.008$; Bonferroni-corrected Wilcoxon test) (S3 Fig).

## Results

Artificial caterpillars encountered arthropod attack on 83% of the plots. At plot level, absence of arthropod attack occurred mainly at low local temperatures, while attack (predation rates > 0) was observed with 80% probability when local mean temperatures were above 7°C (Fig 1, S1 Table). On plots with arthropod attack, on average 26% (mean; ± 19% SD) of the artificial caterpillars were attacked per plot within 2-d exposure; across all plots, the average predation rate was 21% (mean± 20% SD).

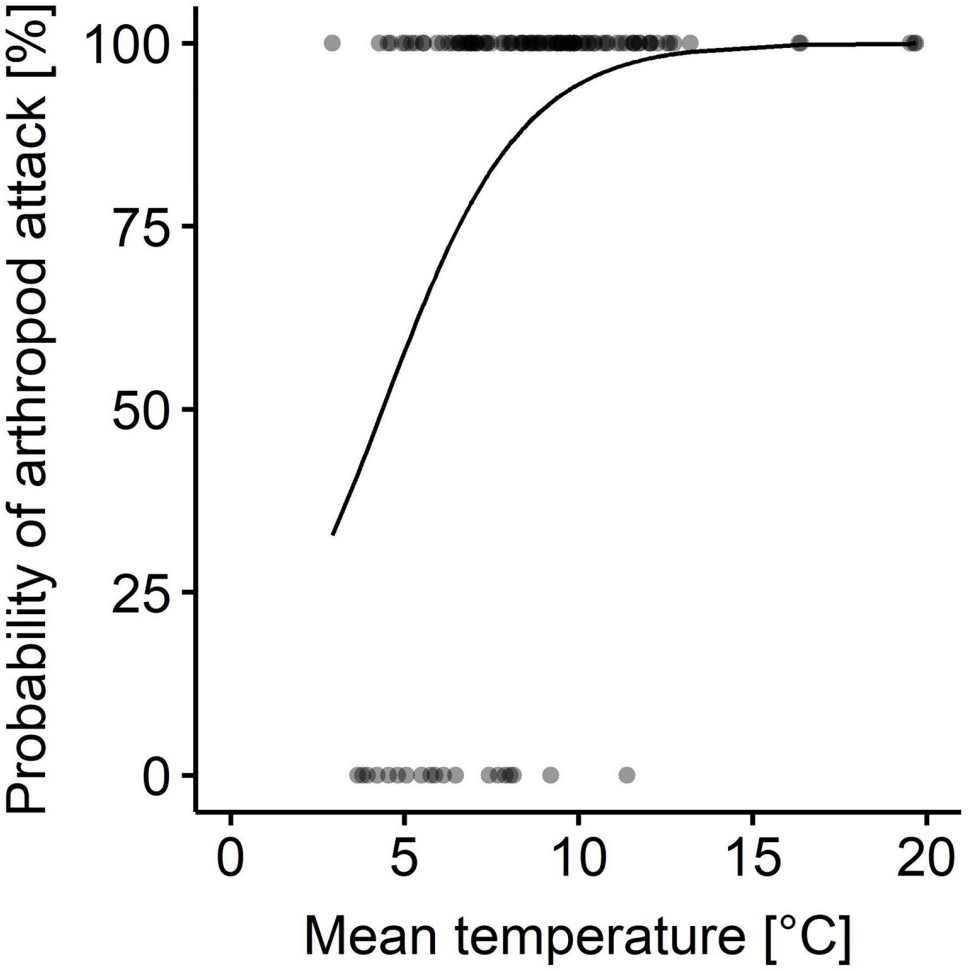

**Fig 1. Probability of arthropod attack relative to local mean temperature during artificial caterpillar exposure.**
Logistic regression curve and dots indicate absence (0) and presence (1) of arthropod attack on artificial caterpillars at plot level.

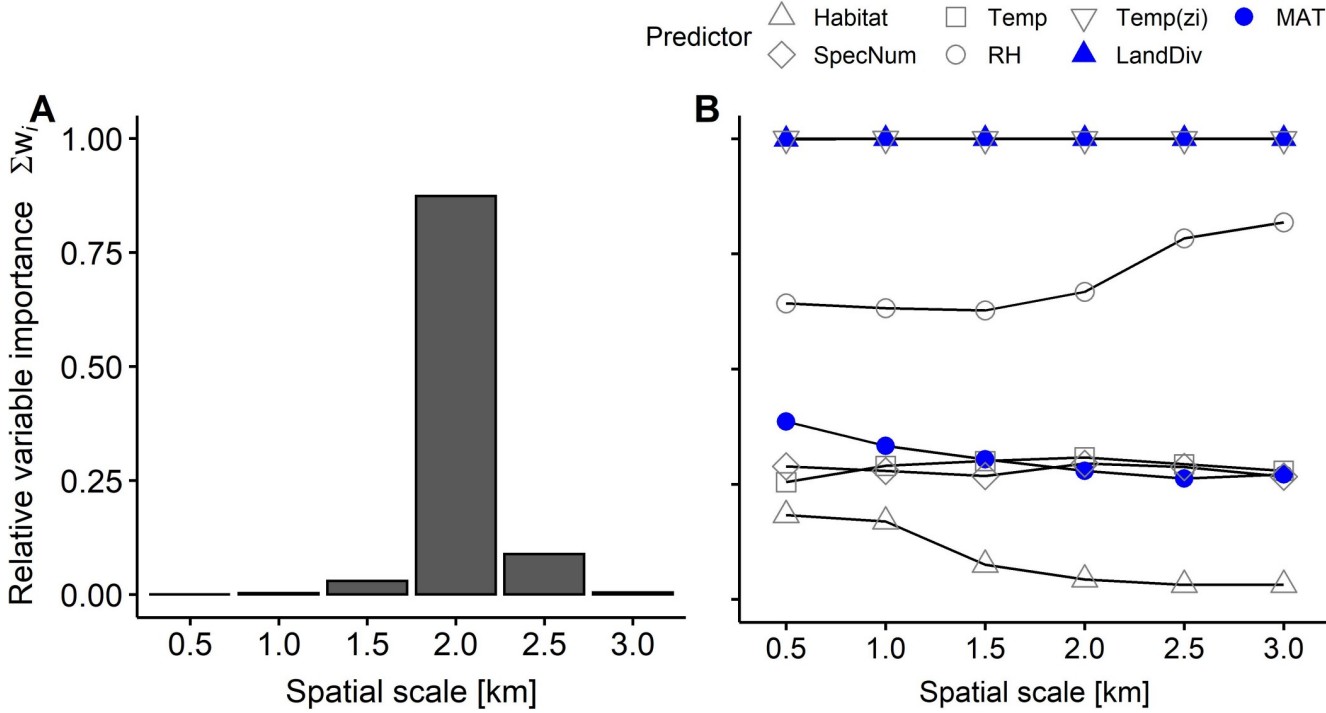

**Fig 2. Relative importance (sum of Akaike weights) for explaining arthropod predation rates of A) spatial scale (i.e. models with all possible predictor combinations at one scale relative to the others) and of B) each candidate predictor per spatial scale.** White symbols refer to habitat characteristics (Habitat: habitat type, SpecNum: plant species richness, Temp or RH: local mean temperature or mean relative humidity during artificial caterpillar exposure, zi: included as zero-inflation term) and filled blue symbols to regional factors (LandDiv: landscape diversity, MAT: multi-annual mean temperature). Landscape diversity is the only landscape parameter (value changes with spatial scale).

Due to landscape diversity as landscape parameter, models at intermediate scales (1.5, 2.0 or 2.5-km)–particularly at 2-km scale–were more important for explaining arthropod predation rates than models at smaller (0.5 km, 1.0 km) or larger scales (3.0 km), as shown by sum of Akaike weights ($\Sigma w_i$, Fig 2A). The relative importance of candidate predictors for explaining arthropod predation rates revealed a similar pattern across all spatial scales, with high relative importance of landscape diversity and local mean temperature as zero-inflation term, intermediate relative importance of mean relative humidity, and low relative importance of MAT, plant species richness, local mean temperature (as fixed effect) and habitat type (Figs 2B and 3). Thus, landscape diversity and–as a zero-inflation term–local mean temperature have a high probability to appear in the best fitting model across spatial scales (Fig 2B), with the most substantial contribution in models including landscape diversity at the intermediate 2-km scale (Fig 2A, see also S3 Table).

Multi-model averaging revealed that, arthropod predation rates were similar among habitat types (Fig 3A; mean ± SD, forests 0.20 ± 0.20, grasslands 0.22 ± 0.20, arable fields 0.21 ± 0.20, settlements 0.21 ± 0.20), and across the observed range of plant species richness (Fig 3B) and local mean temperature (Fig 3C), while higher relative humidity tended to decrease arthropod predation rates (Fig 3D and S3 Table). Local mean temperature as zero-inflation term equals a higher probability of arthropod attack at plot level with higher local mean temperatures (Fig 1). Particularly at 2-km scale (Fig 2A), arthropod predation rates increased towards diverse landscapes (Fig 3E). Higher maximum predation rates and more frequently high predation rates were observed in more diverse landscapes than landscapes dominated by a single land cover type (Fig 3E, e.g. compare landscape diversity < 0.69 and ≥ 0.69, landscape diversity

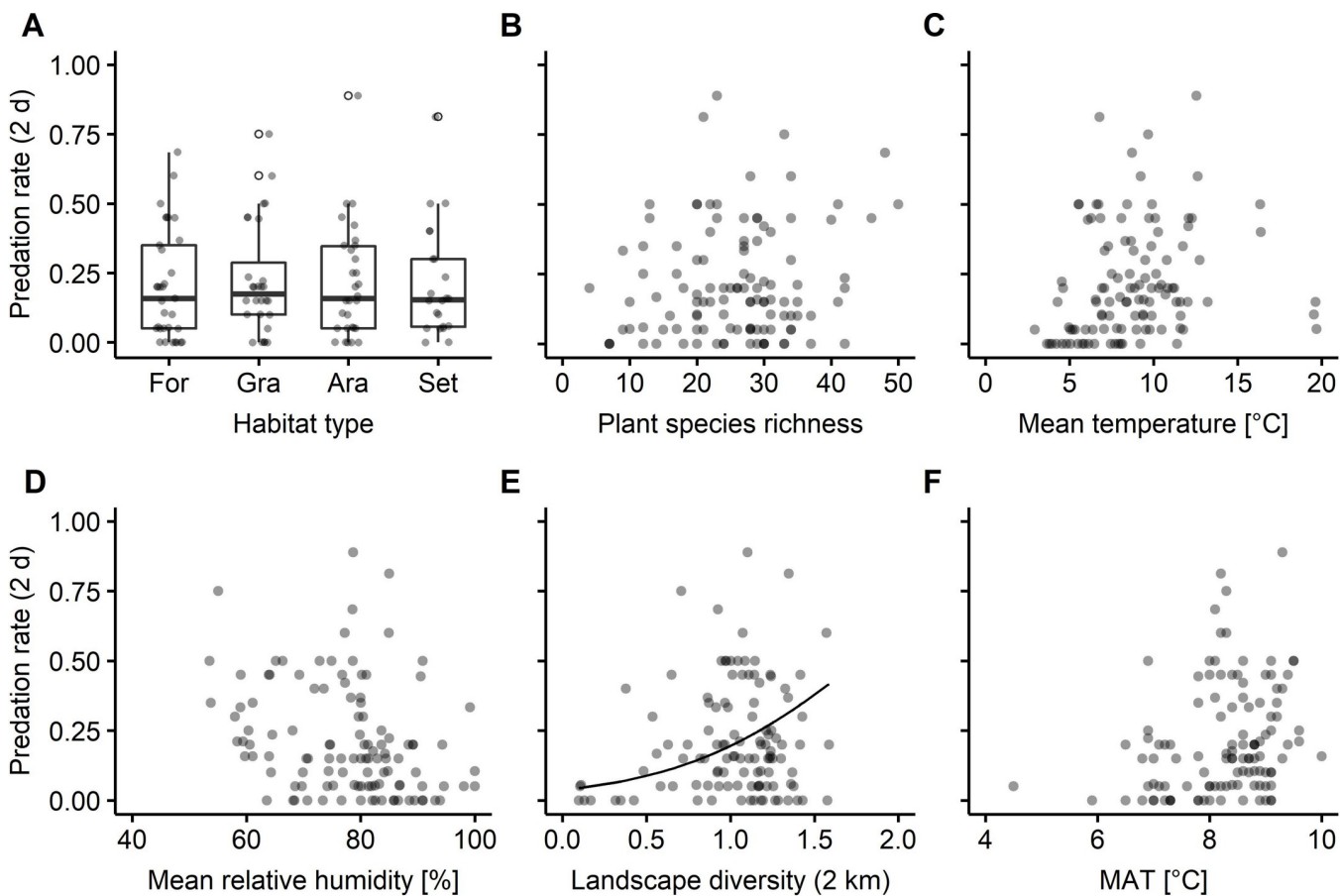

**Fig 3. Relationship between arthropod predation rates and candidate predictors.** A) Habitat type (For: forest, Gra: grassland, Ara: arable field, Set: Settlement), B) plant species richness, C+D) local mean temperature and mean relative humidity during artificial caterpillar exposure, E) landscape diversity at 2-km scale and F) multi-annual mean temperature (MAT). Light grey dots present values per plot; overlapping dots appear darker. In A) circles indicate outliers. In B-F) solid lines indicate model predictions of the best model derived through multimodel averaging.

value of 0.69 equals an effective number of two land-cover types). MAT did not substantially affect arthropod predation rates (Fig 3F). We observed no interaction effects of any predictor on arthropod predation rates depending on habitat type (S4 Table).

## Discussion

In this study, we assessed drivers of arthropod predation in open herbaceous vegetation in typical habitat types of the temperate region. Arthropod predation rates in different habitat types were similar and responded similarly to both local and regional drivers. Towards diverse landscapes, particularly at 2-km scale, arthropod predation rates increased, whereas they tended to decrease towards higher mean relative humidity and were frequently absent from plots with low local mean temperatures. Plant species richness and MAT did not substantially affect arthropod predation rates.

The observed average arthropod predation rate of 21% (in 2 days) in May was in the same order of magnitude as reported in other studies on artificial caterpillars at ground-level in temperate regions, when assuming that arthropod predation rates scale linearly with exposure time [see 38] and tend to increase from spring towards summer [6, 16]. In open herbaceous vegetation, Hertzog et al. [16] obtained average arthropod predation rates of 15% (per day) in

May, and Meyer et al. [38] determined arthropod attack marks on 51% of the recovered artificial caterpillars (after 3 days) in summer.

Among local habitat types (forest, grassland, arable field, settlement) arthropod predation rates were not substantially different and high arthropod predation rates were observed in all habitat types. However, large variation in predation rates among plots of the same habitat type may result from largely different natural enemy communities due to a selective permeability of habitat edges. This permeability depends both on characteristics of the habitat edge–e.g. of natural or anthropogenic origin [39]–and on the behaviour of a predator–e.g. habitat and trophic specialist or generalist [40, 41]. Thus, large variation in spill-over from adjoining habitat into open herbaceous vegetation possibly masked differential effects of local habitat types on arthropod predation rates. This assumption of variation in spill-over from adjoining habitat is supported by the significant impact of landscape diversity on local predation rates.

Towards higher landscape diversity (particularly at 2-km scale), arthropod predation rates increased. Thus, in more diverse landscapes natural enemy communities were likely denser [40, p. 218], richer in the number of species [12] or more frequently included effective predators [12, 13]. However, among plots in diverse landscapes we also observed large variability in predation rates, which may have several reasons. First, natural enemies may respond variably to diverse landscapes depending on i) the presence, proportion and combination of certain land-cover types–which may differ in their supply of complementing or supplementing resources [22, 41]–, ii) the permeability of boundaries between land-cover types–but also at finer scales–[40, 41], and therefore iii) the configuration of land-cover types [42]. Second, changes in natural enemy communities likely mediate landscape-diversity effects on predation rates, but it is not yet fully understood which changes landscape diversity elicits in natural enemy communities [see 40, p. 218], and how and under which conditions this links to altered predation functions [12, 13, 43]. Thus, landscape diversity promotes predation rates, but variability in predation rates in diverse landscapes–and elucidated potential sources of this–point out future research directions to derive more specific recommendations for landscape management aiming to promote top-down regulation of herbivores and potentially also of agricultural pests.

The absence of habitat type effects but increasing predation rates towards higher landscape diversity does not mean that directly adjoining habitat type is less important to arthropod predation than general landscape diversity, yet it suggests that the landscape composition of the intermediate surrounding (i.e. 2-km radius) impacts natural enemy communities in a way that can alter predation rates independent of the dominant habitat in the vicinity. Thus, our results provide first evidence that landscape diversity favours high predation services (in open herbaceous vegetation) across typical habitat types in the temperate region.

Local weather conditions during artificial caterpillar exposure partially shaped arthropod predation. In our study, higher local mean temperatures made it more likely to observe predation (predation rates > 0), but did not substantially increase predation rates. This seems to be in contrast to observations from pitfall trap catches, where numbers of many ground-active arthropod species in the catches increased with temperature [44], which similarly could have increased the likelihood of encounter with an artificial caterpillar. However, as we did not study predation rates as time-series but on different plots, natural enemy communities possibly differed between plots and entailed arthropod species with different temperature preferences [19] and sensitivities [see 44], which can explain the absence of a clear temperature relationship in our study. Furthermore, local mean temperatures measured 1-m above ground possibly reflected the conditions experienced by a predator differently depending on the effects of vegetation structure on microclimate and the daily activity pattern. Thus, local weather conditions may influence predation rates but this effect might be masked in our study, possibly

through differences in natural enemy communities among plots and a discrepancy between the measured and experienced temperature by ground-active arthropods. However, we more frequently observed the absence of attack (predation rates = 0) at low local mean temperatures. Accordingly, temperature thresholds may apply more broadly to arthropod predation, at least in spring. Both, because emergence after overwintering is temperature-dependent [45] and temperature thresholds of relevant predators may not have been reached in colder areas of our study region or not long enough for relevant predators to move onto the plots, and–maybe even more importantly–because initiation of daily activity seems to depend on certain temperatures [19, p. 13]. Thus far only few studies, which quantified predation, have reported on local weather conditions [46]. Our results provide further evidence that local temperature impacts predation and thus should be considered when interpreting predation functions.

MAT did not substantially affect arthropod predation rates. This may have several reasons. First, natural enemy communities were not substantially altered along the observed MAT gradient, or second, different natural enemy communities can provide similar predation rates. Although we cannot test the first reason, studies reporting on increased predation rates towards lower latitudes and altitude [25] or towards higher mean annual temperatures [24] were conducted at a global scale. Thus, various factors may obscure a (weak) climate effect in studies covering a fraction of the global temperature range, such as our study. Indeed, we observed large variation in predation rates among plots of similar MAT, which may suggest that other factors impact natural enemy communities more strongly than MAT. However, even if natural enemy communities change along the MAT gradient, this may not have led to differences in predation rates. This is supported by the observation that several independent studies using artificial caterpillars in temperate regions reported predation rates in the same order of magnitude (see above)–which likely encompasses large differences in natural enemy communities–, but also by the marked relevance of key predators for predation functions, e.g. compared to natural enemy richness [12]. Key predators can be, for instance, particular voracious predator species [47] and predators with specific feeding traits matching the 'vulnerability trait' of the prey [48]. Thus, high predation rates across the covered MAT gradient illustrate the potential to increase predation rates (e.g. through landscape management) independent of a potential moderate increase of MAT in the near future.

Plant species richness did not substantially affect arthropod predation rates in our study, whereas Barnes et al. [15] reported increasing top-down control and Hertzog et al. [16] increasing invertebrate predation rates towards higher plant species richness. However, these positive effects of plant species richness on predation rates were reported from grassland experiments [15, 16], whereas we report from a multi-scale field experiment. This likely included much more variation in natural enemy communities and also considerable differences in plant species pools between plots. Thus, plant species richness may indirectly affect arthropod predation rates depending on the natural enemy community composition and the plant species pool, but our data suggests that this is not a ubiquitous or dominant pattern.

Our results provide insights into herbivore regulation through arthropod predators, but are limited by the method of artificial caterpillars as sentinel prey. Common predators on artificial caterpillars at ground-level are chewing insects, especially carabids [6]. Properties of the artificial caterpillars such as length [49] and colour [50] act as a filter–with yet unknown specificity–on the interacting predators. Furthermore, predation rates on artificial caterpillars do not directly translate into successful predation attempts as the complexity of predator-prey interactions is reduced by e.g. prey mobility [51] and defensive traits such as cuticular toughness [48]. However, predation rates on artificial caterpillars are widely recognised for their standardisable estimate [27, 52] and can provide unique insights into drivers of predation functions through generalist predators, which are otherwise impossible to obtain.

## Conclusion

We conclude that landscape diversity favours high arthropod predation rates in open herbaceous vegetation across typical habitat types in the temperate region, while adjoining habitat type and plant species richness are of minor importance when studying a large spatial extent with possibly vastly different natural enemy communities. However, more research is needed on the underlying mechanisms of the landscape diversity effect to deduce more specific management options for improved top-down control of herbivores, and for enhanced natural pest control in agricultural ecosystems. Besides, local weather conditions impact predation, e.g. low local mean temperatures can limit predation, and hence should be considered when interpreting predation rates. With respect to MAT, arthropod predation rates did not substantially change and high rates were observed across the covered climatic gradient, which highlights the potential to increase predation rates (e.g. through landscape management) independent of potential moderate temperature increases in the near future.

## Supporting information

**S1 Table. Model output on the probability of arthropod attack on plot level.** (presence-absence of attack from a plot, binomial generalized linear mixed model) including local mean temperature during artificial caterpillar exposure as predictor compared to an empty model (null, null model). Bold font highlights the best model based on ΔAICc < 2 and parsimony.
(PDF)

**S2 Table. Predictor variable details and Pearson's correlation coefficients.** Included in multimodel averaging on arthropod predation rate models based on 113 study sites. Significant correlations based on α = 0.05 are indicated as following: $P < 0.05^*$, $P < 0.01^{**}$, $P < 0.001^{***}$.
(PDF)

**S3 Table. Model output of arthropod predation rate models.** (zero-inflated binomial generalized linear mixed model) with different parametrization (best: best model based on ΔAICc < 2 and parsimony; ΔAICc < 4; full: model containing all candidate predictors; null: null model with or without temperature as zero-inflation term) at the best spatial scale identified by multimodel averaging (2-km scale). Best model is highlighted in bold font.
(PDF)

**S4 Table. Model output of arthropod predation rate models including interactive effects with habitat type.** (zero-inflated binomial generalized linear mixed model) at the best spatial scales identified by multimodel averaging (2-km scale). Interaction terms are added to the original best model and to null models containing only temperature during exposure as zero-inflation term. Asterisks between candidate predictors indicate that both main effects and their interaction term is included. Best model parametrization is derived based on ΔAICc < 2 and parsimony. Best models are highlighted in bold font.
(PDF)

**S1 Fig. Maps showing study regions within Bavaria, Germany (left), and plots within an example region (right).** Squares indicate locations of study regions. Different colours represent the 15 combinations of climate zones (1–5: multi-annual mean temperature from 1981–2010; < 7˚C, in 0.5˚C steps to 9˚C, > 9˚C) and regional land-use types (nature = near-natural, agriculture and urban), in four replicates. Regional land-use types were defined as near-natural when > 85% of the region were covered by near-natural vegetation with a minimum of 50% forest, as agriculture when > 40% were covered by arable land and managed grassland, and as urban when > 14% were covered by housing, industry and traffic infrastructure. The land

cover map, to the right, shows six main land use types (different colours), three plot locations marked by "x" within the dominating land use types of the region, and 1-km "buffer zones" around the plots.
(PDF)

**S2 Fig. Pearson correlation between arthropod predation rates assessed in the field and in the lab.** Dots indicate values per plot; overlapping dots appear darker. The dashed grey line presents a hypothetically perfect correlation (r = 1) and the solid black line, the observed correlation based on α = 0.05, P < 0.001***.
(PDF)

**S3 Fig. Relationship between habitat type and other candidate predictors of arthropod predation rates.** Dots indicate values per plot; overlapping dots appear darker. Asterisks highlight significance levels of P < 0.05* and P < 0.01**. Letters indicate significant differences between habitat types based on Bonferroni-corrected pairwise comparisons using Wilcoxon rank sum test.
(PDF)

## Acknowledgments

We are grateful to the landowners, leaseholders, municipalities and the Bavarian State Forestry, who facilitated this project. Special thanks go to Bastian Häfner and Paul Geisendörfer for preparing the artificial caterpillars. We thank the Bavarian Office for Surveying and Geographic Information, the European Environment Agency of the European Union under the framework of the Copernicus programme and the Bavarian State Ministry of Agriculture and Forestry for providing ATKIS 2019, CORINE 2018 and IACS 2019 land cover data, respectively. This study was conducted within the framework of the joint project *LandKlif* (https://www.landklif.biozentrum.uni-wuerzburg.de/).

## Author Contributions

**Conceptualization:** Ute Fricke, Ingolf Steffan-Dewenter, Jie Zhang, Sarah Redlich.

**Funding acquisition:** Ingolf Steffan-Dewenter.

**Investigation:** Ute Fricke, Jie Zhang, Cynthia Tobisch, Sandra Rojas-Botero, Caryl S. Benjamin, Jana Englmeier, Cristina Ganuza, Maria Haensel, Rebekka Riebl, Johannes Uhler, Lars Uphus.

**Methodology:** Ute Fricke, Ingolf Steffan-Dewenter, Cynthia Tobisch, Sandra Rojas-Botero, Jörg Ewald, Johannes Kollmann, Sarah Redlich.

**Supervision:** Ingolf Steffan-Dewenter, Sarah Redlich.

**Writing – original draft:** Ute Fricke.

**Writing – review & editing:** Ute Fricke, Ingolf Steffan-Dewenter, Jie Zhang, Cynthia Tobisch, Sandra Rojas-Botero, Caryl S. Benjamin, Jana Englmeier, Cristina Ganuza, Maria Haensel, Rebekka Riebl, Johannes Uhler, Lars Uphus, Jörg Ewald, Johannes Kollmann, Sarah Redlich.

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
