## [Decision Letter · Decision Letter 0]

28 Mar 2022

PONE-D-22-04964Landscape diversity and local temperature, but not climate, affect arthropod predation among habitat typesPLOS ONE

Dear Dr. Fricke,

Thank you for submitting your manuscript to PLOS ONE. After careful consideration, we feel that it has merit but does not fully meet PLOS ONE’s publication criteria as it currently stands. Therefore, we invite you to submit a revised version of the manuscript that addresses the points raised during the review process.

Your manuscript is very interesting and easy to read. We suggest you to make minor revision and please carefully consider reviewers suggestions.

We look forward to receiving your revised manuscript.

Kind regards,

Eve Veromann, PhD

Academic Editor

PLOS ONE

Journal Requirements:

Reviewers' comments:

Reviewer's Responses to Questions

**Comments to the Author**

1. Is the manuscript technically sound, and do the data support the conclusions?

Reviewer #1: Yes

Reviewer #2: Yes

2. Has the statistical analysis been performed appropriately and rigorously? 

Reviewer #1: Yes

Reviewer #2: Yes

3. Have the authors made all data underlying the findings in their manuscript fully available?

Reviewer #1: Yes

Reviewer #2: No

4. Is the manuscript presented in an intelligible fashion and written in standard English?

Reviewer #1: Yes

Reviewer #2: Yes

5. Review Comments to the Author

Reviewer #1: Dear Editor Prof. Eve Veromann,

Dear Authors,

please find my comments on the paper “Landscape diversity and local temperature, but not climate, affect arthropod predation among habitat types”

The authors showed a very well done and intriguing piece of research about arthropod predation in temperate areas. They correctly pointed out the innovation their research fulfilled: a rare piece of research which study, citing the paper, “the combined effects of local habitat type, plant species richness, weather, landscape diversity and multi-annual mean temperature on arthropod predation”. These data, even if limited in the continental European region, are very interesting to understand the amount of predation rates we nowadays can expect in a vast portion of Europe, and their results, as they pointed out, can help to understand what the potential is to increase these rates through landscape management.

In the introduction, the authors reported concisely, but logically and exhaustive, the context of the research on arthropod predation, as well as open research question and we can clearly read the logic behind the experiment layout we’re going to read in the M&M section.

They measured arthropod predation in an impressive series of plot (147 out of 179 plots, which they reduce, correctly, to 113 after a critic revision of their dataset) in the Bavaria region, Germany. The methodologies they uses are correct, as well as site selection criteria are almost clear and reasonable.

Data analysis have been done correctly, deeply, and using modern techniques, programs, and rationale. I would like to compliment with the authors, because this is one of the best examples of the use of statistics to achieve comprehensive results I recently found.

The authors analysed the results in a logic way. They read their results formulating clear hypotheses fond on recent and appropriate research, but at the same time they were very honest about the limits of their findings. In addition, they correctly criticized the validity of their finding because of plasticine predation limits (like any other experiment of this kind in field, I would say).

Thus, my suggestion to Prof. Veromann is to accept the paper with (very) minor revision. Here I report line by line comments.

MATERIAL AND METHODS

Line 90. I’m missing here the amount (and rational) of climate zones. I suspect they’re 5 climatic zones X 3 landscape-scale land use types = 15 combinations. Are the climatic zones from the reference n° 30? I tried to check it, but it is not so user-friendly. Please explain how, of the source of climatic zones.

Reviewer #2: In this manuscript, Fricke and colleagues used an innovative multi-scale sampling design to disentangle the effects of multiple drivers on arthropod predation rate. Specifically, they tested how arthropod predation rate varied in contrasting habitats across climate and land-use gradients. I am very enthusiastic about the novel sampling design adopted by the authors, although some additional information describing in more detail the selection and the characteristics of the selected ‘regions’ and ‘plots’ would be beneficial (see specific comments below). The manuscript is well written and a pleasure to read. This allowed the manuscript to be quickly and easily understood. Although the results are in any case interesting, I would have expected, especially considering the sampling design, a major role of climate and a possible interactive effect between climate and landscape on this important function. I am wondering if this could be related to the sampling approach (just a unique sampling date) adopted by the authors. Anyway, this work is certainly of interest, but there are still several minor questions that the authors should address. Hope the authors find my comments helpful.

SPECIFIC COMMENTS:

L59 It is not clear what you mean by ‘correlation of the latter’.

L81 I would spend a few more words why you used this method to measure arthropod predation rate. It’s clear reading the Method section, but I think it is also important to introduce it earlier.

L89-91 The sampling design seems very interesting, but I would like to know something more how the selection was done and what are the characteristics of each combination (e.g. climate, landscape, etc.). Also a figure representing the sampling design might be useful.

L93 How did you define typical for the respective region? Did you use a specific threshold?

L94 Also in forests?

L115 Just one sampling per plots? Right?

L119-121 I understand the reasoning of that, but there is a risk that the possible habitat effect can be masked in this specific case (predation rate measured using green artificial caterpillars and mostly determined by ground-dwelling predators). The herbaceous vegetation, regardless of the habitat where is embedded, has the same effect.

L 148 Would it be possible to make the code used open?

L 258-261 In my opinion, what you are observing here is not a habitat effect per se, as you sampled an open herbaceous vegetation in all habitats. The micro habitat conditions (herbaceous vegetation) are more important than the habitat in se. This is might particular true for ground-dwelling predators.

L 283 This is true Just in part according to your results and what you wrote below.

L305 What do you mean with ‘long-term temperature’? Not clear to me!

L344 Why only for agricultural ecosystems?

6. PLOS authors have the option to publish the peer review history of their article (what does this mean?). If published, this will include your full peer review and any attached files.

Reviewer #1: No

Reviewer #2: No

---

## [Author Response · Author response to Decision Letter 0]

4 Apr 2022

For the response to the editor and the reviewers, please see attached 'Response to Reviewers'.

---

## [Editor Report · Decision Letter 1]

13 Apr 2022

Landscape diversity and local temperature, but not climate, affect arthropod predation among habitat types

PONE-D-22-04964R1

Dear Dr. Fricke,

We’re pleased to inform you that your manuscript has been judged scientifically suitable for publication and will be formally accepted for publication once it meets all outstanding technical requirements.

Kind regards,

Eve Veromann, PhD

Academic Editor

PLOS ONE
---

## [Editor Report · Acceptance letter]

21 Apr 2022

PONE-D-22-04964R1 

Landscape diversity and local temperature, but not climate, affect arthropod predation among habitat types 

Dear Dr. Fricke:

I'm pleased to inform you that your manuscript has been deemed suitable for publication in PLOS ONE. Congratulations! Your manuscript is now with our production department. 

Kind regards, 

on behalf of

Dr. Eve Veromann 

Academic Editor

PLOS ONE